# Numerical Study of Sub-Gap Density of States Dependent Electrical Characteristics in Amorphous In-Ga-Zn-O Thin-Film Transistors

**Do-Kyung Kim [1], Jihwan Park [1], Xue Zhang [2], Jaehoon Park [3,\*] and Jin-Hyuk Bae [1,4,\*]**

[1] School of Electronic and Electrical Engineering, Kyungpook National University, 80 Daehakro, Bukgu, Daegu 41566, Korea; kdk7362@naver.com (D.-K.K.); nrnr14@naver.com (J.P.)

[2] College of Ocean Science and Engineering, Shandong University of Science and Technology, Qingdao 266590, China; zhangxue00@sdust.edu.cn

[3] Department of Electronic Engineering, Hallym University, Chuncheon 24252, Korea

[4] School of Electronics Engineering, Kyungpook National University, 80 Daehakro, Bukgu, Daegu 41566, Korea

\* Correspondence: jaypark@hallym.ac.kr (J.P.); jhbae@ee.knu.ac.kr (J.-H.B.); Tel.: +82-33-247-2524 (J.P.); +82-53-950-7222 (J.-H.B.)

**Abstract:** We demonstrate the effect of the sub-gap density of states (DOS) on electrical characteristics in amorphous indium-gallium-zinc (IGZO) thin-film transistors (TFTs). Numerical analysis based on a two-dimensional device simulator Atlas controlled the sub-gap DOS parameters such as tail acceptor-like states, tail donor-like states, Gauss acceptor-like states, and Gauss donor-like states in amorphous IGZO TFTs. We confirm accuracy by exploiting physical factors, such as oxygen vacancy, peroxide, hydrogen complex, band-to-band tunneling, and trap-assisted tunneling. Consequently, the principal electrical parameters, such as the threshold voltage, saturation mobility, sub-threshold swing, and on-off current ratio, are effectively tuned by controlling sub-gap DOS distribution in a-IGZO TFTs.

**Keywords:** In-Ga-Zn-O; thin-film transistors; sub-gap density of states; electrical characteristics; numerical analysis

## 1. Introduction

Thin-film transistors (TFTs) have received significant attention because of the rapid development of displays, sensors, computing, radiofrequency tags, and analog signal processing [1–6]. Specifically, TFTs based on metal-oxide, organic, and low-dimensional materials have advanced, achieving high electrical performances, optical transparency, and mechanical flexibility [7–10]. Among the various next-generation material-based TFTs, amorphous oxide semiconductor (AOS) TFTs based on multi-components exhibit high uniformity because of their high mobility, amorphous phase, and high transparency and flexibility [11–13]. Indium-gallium-zinc (IGZO) TFTs are the most representative AOS TFTs because of the advantages of carrier concentration controllability, electrical characteristics stability, and process compatibility with present fabrication [14–16]. A challenging issue in IGZO TFTs is enhancing electrical performance to drive high-quality active-matrix electronics, such as organic light-emitting diodes, micro light-emitting diodes, and high-resolution displays and sensors. To enhance the electrical performance of IGZO TFTs, many studies on IGZO TFTs exist, and oxygen vacancy (OV), peroxide (PO), and hydrogen complex (HC) models have been suggested [17–20]. However, progress has been insufficient to fully understand the change of electrical characteristics induced by the sub-gap density of states (DOS), even though sub-gab DOS distributions are highly related to the electrical characteristics of AOS TFTs.

In this study, we controlled sub-gap DOS distribution in variables such as tail acceptor-like states ($N_{TA}$), tail donor-like states ($N_{TD}$), Gauss donor-like states ($N_{GA}$), and Gauss acceptor-like states ($N_{GD}$). Therefore, we use Silvaco's two-dimensional device simulator Atlas to examine the effect of sub-gab DOS distributions on electrical characteristics. Consequently, the results show that the characteristics of electrical parameters, such as the threshold voltage ($V_{th}$), saturation mobility ($\mu_{sat}$), sub-threshold swing (S), and on-off current ratio ($I_{on}/I_{off}$), significantly affect the sub-gap DOS distribution.

## 2. Simulation Methodology

For the AOS IGZO TFT simulation, top-contact bottom-gate staggered structure was used in this study (Figure 1a). In the simulation, indium tin oxide (ITO) and aluminum (Al) were used for the gate and the source and drain electrodes, respectively. The work functions of ITO and Al are 4.7 eV and 4.33 eV, respectively. The lengths of both source and drain are 1 μm, and the gate length is 10 μm. The thickness of the electrodes is 50 nm. The gate insulator is a 100-nm-thick $SiO_2$, and the active layer material is IGZO. The channel width and length are 100 μm and 10 μm, respectively. The affinity and bandgap of the IGZO parameters are 4.16 eV and 3.05 eV, respectively, and the conduction band ($N_C$) and valence band ($N_V$) effective DOS are calculated using Equations (1) and (2).

$$N_C = 2\left(\frac{2\pi M_C kt}{h^2}\right)^{\frac{3}{2}},$$ (1)

$$N_V = 2\left(\frac{2\pi M_V kt}{h^2}\right)^{\frac{3}{2}},$$ (2)

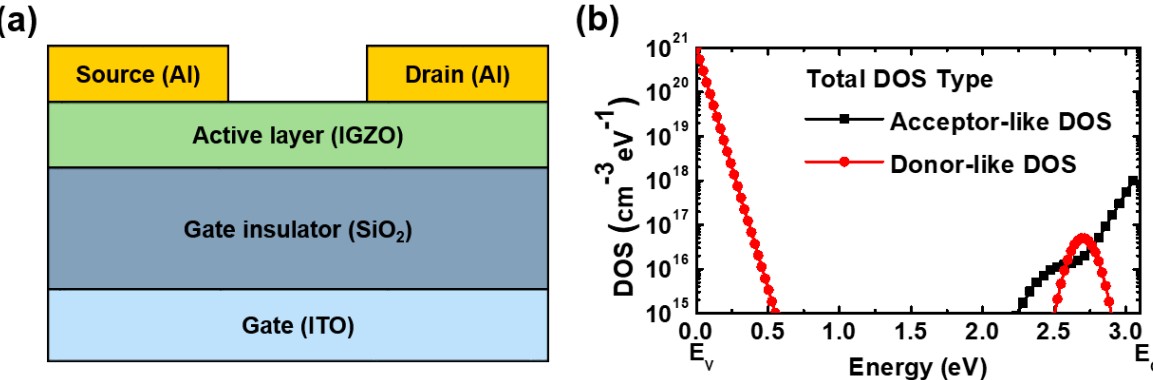

**Figure 1.** (**a**) Schematic cross-sectional view of the structure of the bottom-gate staggered structure of the device used in this work. (**b**) Density of states (DOS) is shown as a function of energy (E). The black squares are total acceptor-like traps sub-gap DOS, and the red circles are total donor-like traps sub-gap DOS.

In this study, we assumed that the effective mass of the electron and hole are 0.34 $M_O$ and 21 $M_O$ [21]. The values calculated using Equations (1) and (2) are $5 \times 10^{18}$ cm$^{-3}$ eV$^{-1}$ and $2.4 \times 10^{21}$ cm$^{-3}$ eV$^{-1}$, respectively. This study assumed that localized states and $N_{GD}$ are equal. We also assumed that the electron carrier concentration is partially ionized from $N_{GA}$ using the OV model. Figure 1b presents the reference sub-gap DOS consisting of $N_{TA}$, $N_{TD}$, $N_{GA}$, and $N_{GD}$. Each formula follows the sub-gap DOS model [21]. The $N_{TA}$ distribution equation ($G_{TA}$ (E)) in the sub-gap DOS is (3).

$$G_{TA}(E) = N_{TA}e^{\frac{(E-E_C)}{W_{TA}}},$$ (3)

where $E$ is the electron energy, $E_C$ is the conduction band edge energy, $N_{TA}$ is the sub-gap DOS at $E-E_C$, and $W_{TA}$ is the characteristic decay energy. The $N_{TD}$ distribution equation ($G_{TD}$ (E)) in the sub-gap DOS is given by Equation (4).

$$G_{TD}(E) = N_{TD}e^{\frac{(E_V-E)}{W_{TD}}}, \tag{4}$$

where $E_V$ is the valence band edge energy, $N_{TD}$ is the sub-gap DOS at $E-E_V$, and $W_{TD}$ is the characteristic decay energy. The $N_{GA}$ distribution equation ($G_{GA}$ (E)) in the sub-gap DOS is given by Equation (5).

$$G_{GA}(E) = N_{GA}e^{[-(\frac{(E-E_{GA})}{W_{GA}})^2]}, \tag{5}$$

where $E_{GA}$ is the $N_{GA}$ energy peak, $N_{GA}$ is the Gaussian acceptor-like states at $E_{GA}-E$ in the sub-gap DOS, and $W_{GA}$ is the characteristic decay energy. The $N_{GD}$ distribution equation ($G_{GD}$ (E)) in the sub-gap DOS is given by Equation (6).

$$G_{GD}(E) = N_{GD}e^{[-(\frac{(E-E_{GD})}{W_{GD}})^2]}, \tag{6}$$

where $E_{GD}$ is the $N_{GD}$ energy peak, $N_{GD}$ is Gaussian donor-like states at $E-E_{GD}$ in the sub-gap DOS, and $W_{GD}$ is the characteristic decay energy. In the semiconductor based on Si and compound case, $N_{GD}$ is located near $E_V$, but in the AOS case, $N_{GD}$ is positioned near $E_C$ because the $N_{GD}$ position changes the Madelung potential effect of the OV model [17].

The OV model demonstrates the $N_{GD}$ moves from a deep trap state to localized states because of the Madelung potential. Furthermore, the electrons are generated by removing oxygen bonding. The OV model is associated with Equation (7).

$$V_O{}^X \rightarrow V_O{}^{+2} + 2e, \tag{7}$$

where the neutrality charge is $X$, the two-plus charge is $+2$, the OV state is $V_O$, and $e$ is the electron. Therefore, the $V_O$ state level is changed by filling or discharging electrons using the Madelung potential. Therefore, $N_{GD}$ would change from deep trap states to localized states. Furthermore, the PO model shows that excess oxygen induces an O–O bonding and identifies changed localized state levels. Thus, the $N_{GD}$ energy level should change in the localized state regime under the PO model [18]. Furthermore, the HC model illustrates that coupling metal atoms and light hydrogen atoms induces various energy levels [19]. Consequently, coupling at various energy levels would change the sub-gap DOS and Fermi level. Therefore, these variable models should demonstrate varying sub-gaps DOS by chemical atomic bonding or breaking. Thus, various parameters in the sub-gap DOS using the models should be applied in the simulation to observe the changes in the electrical characteristics in this study. Table 1 shows other parameters required for simulation. For accurate simulation, the generation-recombination mechanisms tarp-assisted tunneling (TAT) associated with band-to-band (BBT) and Pool–Frenkel Barrier Lowering (PFBL) models must be used [21,22]. The BBT and TAT models [23] should change the generation rate.

$$G_{BBT}(F) = \frac{q^2F^2m_r^{\frac{1}{2}}}{18\pi\hbar E_G^{\frac{1}{2}}}e^{(-\frac{\pi m_r^{\frac{1}{2}}E_G^{\frac{3}{2}}}{2\hbar qF})}, \tag{8}$$

$F$, $m_r$, $q$, and $E_G$ are the local electric field, conduction band effective mass, elementary charge, and bandgap energy, respectively.

**Table 1.** Simulation parameters for the a-indium-gallium-zinc (IGZO) thin-film transistors (TFTs).

| Parameter | Value | Unit | Description |
|---|---|---|---|
| $\mu_n$ | 15 | cm$^2$ V$^{-1}$·s$^{-1}$ | Electron mobility |
| $\mu_p$ | 0.1 | cm$^2$ V$^{-1}$·s$^{-1}$ | Hole mobility |
| $N_{TA}$ | $1 \times 10^{18}$ | cm$^{-3}$ eV$^{-1}$ | Density of tail states at conduction band |
| $N_{TD}$ | $1 \times 10^{20}$ | cm$^{-3}$ eV$^{-1}$ | Density of tail states at valence band |
| $N_{GD}$ | $5 \times 10^{16}$ | cm$^{-3}$ eV$^{-1}$ | Density of Gauss donor-like states |
| $N_{GA}$ | $5 \times 10^{16}$ | cm$^{-3}$ eV$^{-1}$ | Density of Gauss acceptor-like states |
| $W_{TA}$ | 0.08 | eV | Characteristic decay energy of acceptor-like tail states |
| $W_{TD}$ | 0.06 | eV | Characteristic decay energy of donor-like tail states |
| $W_{GD}$ | 0.1 | eV | Characteristic decay energy of donor-like Gauss states |
| $W_{GA}$ | 0.2 | eV | Characteristic decay energy of acceptor-like Gauss states |
| $E_{GD}$ | 2.7 | eV | Energy corresponding to the peak for donor-like Gauss states |
| $E_{GA}$ | 0.5 | eV | Energy corresponding to the peak for acceptor-like Gauss states |
| $N_C$ | $5 \times 10^{18}$ | cm$^{-3}$ eV$^{-1}$ | Effective density of states for conduction band |
| $N_V$ | $2.4 \times 10^{21}$ | cm$^{-3}$ eV$^{-1}$ | Effective density of states for valance band |
| $E_G$ | 3.05 | eV | Energy band |
| $X_e$ | 4.16 | eV | Electron affinity |
| $M_C$ | $0.34\,M_o$ | Kg | Effective mass of conduction band |
| $M_V$ | $21\,M_o$ | Kg | Effective mass of valence band |
| $\varepsilon_s$ | $10\varepsilon_o$ | Fm$^{-1}$ | Dielectric constant |
| $n$ | $10^{15}$ | cm$^{-3}$ | Electron carrier concentration |

## 3. Results and Discussion

Figure 2a shows the $N_{TA}$ versus energy (E). The $N_{TA}$ varies from $1 \times 10^{18}$ to $5 \times 10^{19}$ cm$^{-3}$ eV$^{-1}$ by five intervals. Figure 2b shows that the transfer curve is changing with the $N_{TA}$. It presents the log scaled $I_D$–$V_G$ curve by sweeping $V_G$ from −20 V to 40 V at $V_D = 40$ V of a-IGZO TFTs from each $N_{TA}$. Thus, electrical performance such as low $I_{on}/I_{off}$ from $5 \times 10^{12}$ to $2.27 \times 10^{12}$ A and low $\mu_{sat}$ from 11–7.9 cm$^2$ V$^{-1}$·s$^{-1}$, high $V_{th}$ from 0.35–4.5 V, and high S from 128–438 mV dec$^{-1}$ would be worse. We extract electrical parameters by analyzing the transfer curves. We take advantage of specific equations to extract parameters. First, $I_D$ is given by Equation (6) at the saturation region in the $I_D$–$V_G$ transfer curve.

$$I_D = \mu_{sat} \cdot C_{ox}\left(\frac{W}{2L}\right)(V_G - V_{th})^2, \tag{9}$$

where $W$ is the device width, $L$ is the channel length, and $C_{ox}$ is the semiconductor insulator. Second, the $G_{rad.max}$ of the maximum gradient concept is used to extract $\mu_{sat}$ and $V_{th}$ at the saturation region in the $\sqrt{I_D}$–$V_G$ transfer curve.

$$G_{rad} = \frac{a\sqrt{I_D}}{aV_G} = \left(\mu_{sat} \cdot C_{ox}\left(\frac{W}{2L}\right)\right)^{\frac{1}{2}}. \tag{10}$$

$G_{rad.max}$ is the maximum gradient value. Furthermore, to evaluate the device performance, the $I_{on}/I_{off}$ is critical at the saturation region in the $I_D$–$V_G$ transfer curve.

$$\frac{I_{on}}{I_{off}} = \frac{I_D|V_G = 40V}{I_D|V_G = -20V}. \tag{11}$$

Third, $\mu_{sat}$ is critical for evaluating device performance in the $I_D$–$V_G$ transfer curve at the saturation region in the $\sqrt{I_D}$–$V_G$ transfer curve.

$$\mu_{sat} = \left(\frac{1}{C_{ox}}\right)\left(\frac{2L}{W}\right)(G_{rad.max})^2. \tag{12}$$

Fourth, $V_{th}$ is also necessary for evaluating device performance in the $I_D–V_G$ transfer curve at the saturation region in the $\sqrt{I_D}–V_G$ transfer curve.

$$V_{th} = V_G - \frac{\sqrt{I_D}}{G_{rad.max}}. \tag{13}$$

Fifth, $S$ is concerned with the low power device parameter at the saturation region in the $I_D–V_G$ transfer curve.

$$S = \frac{aV_G}{a\log(I_D)} = \frac{1}{\mu_{sat} \cdot C_{ox}\left(\frac{W}{2L}\right)}. \tag{14}$$

Finally, the concept of $V_{on}$ should be used to evaluate the changing electrical property exactly at the saturation region in the $I_D–V_G$ transfer curve. Note that these phenomena are matched to the multiple trapping and release (MTR) model [24]. Thus, many $N_{TA}$ would interfere with the path of the electron. We elucidate the reasons by investigating the electron concentration distribution in the channel (Figure 2c). The results show that $N_{TA}$ is associated with lowering the electron concentration from top to bottom in the channel because of trapping electrons. Therefore, the $N_{TA}$ would be disturbed to move electrons using the MTR effect. Furthermore, we also ran the simulation in relation to $W_{TA}$. The variation of $W_{TA}$ is 0.08–0.14 eV at $N_{TA} = 10^{18}$ cm$^{-3}$ eV$^{-1}$. However, no change in electrical properties was observed in this study. Therefore, $N_{TA}$ is more important than $W_{TD}$ regarding its effect on electrical performance.

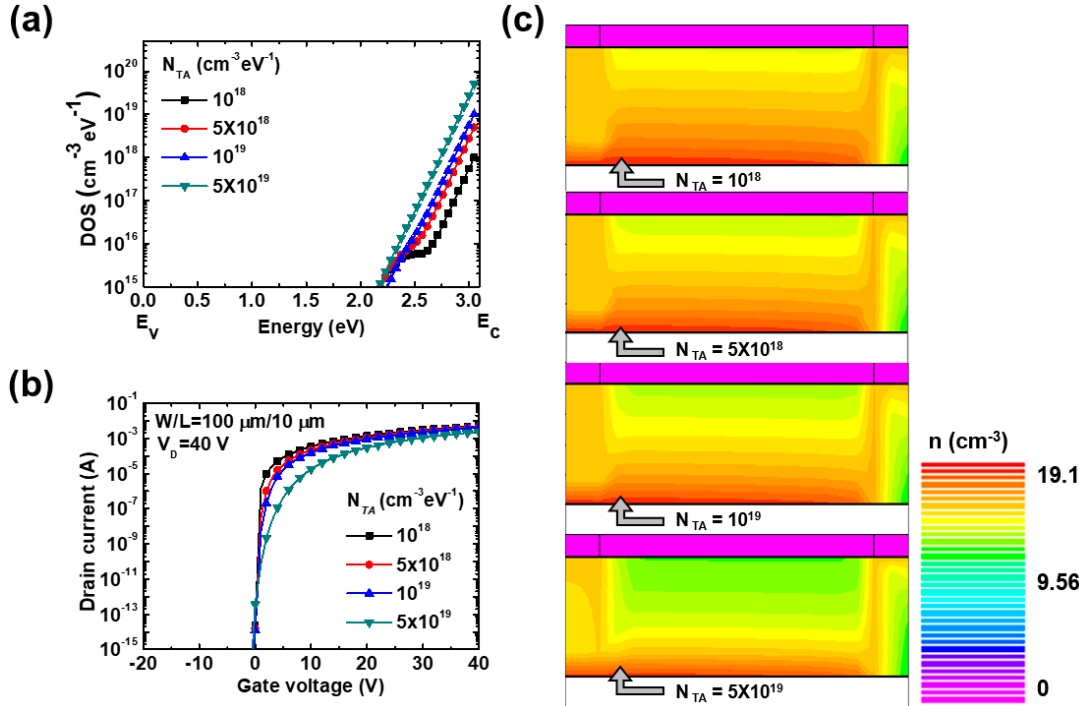

**Figure 2.** (**a**) Density of states (DOS) in sub-gap is shown as a function of energy. The tail acceptor-like states ($N_{TA}$) are controlled from $10^{18}$ to $5 \times 10^{19}$ cm$^{-3}$ eV$^{-1}$. (**b**) Drain current according to the applied gate voltage is shown at 40 V of drain voltage. (**c**) Each layer depicts the electron concentration distribution from top to bottom of $N_{TA}$ in the range of $10^{18}$ to $5 \times 10^{19}$ cm$^{-3}$ eV$^{-1}$ at the saturation region 40 V of gate voltage, and 40 V of drain voltage. The color table displays the electron concentration variations of $0–10^{19.1}$ in varying colors.

Figure 3a shows the $N_{GA}–E$. The variation of $N_{GA}$ is from $10^{16}$ to $5 \times 10^{17}$ by five intervals. Figure 3b shows that the transfer curve is changing with $N_{GA}$. It presents the log scaled $I_D–V_G$ curve by sweeping $V_G$ from $-20$ V–40 V at $V_D = 40$ V of IGZO TFTs from each $N_{GA}$. Therefore, electrical performance,

such as low $I_{on}/I_{off}$ from $4 \times 10^{12}$ to $2.93 \times 10^{12}$ A and low $\mu_{sat}$ from 10.4 to 9.8 cm$^2$ V$^{-1}$·s$^{-1}$, high $V_{th}$ from 0.5 to 2.5 V, high $V_{on}$ from 0.5 to 2.5 V, and high $S$ from 128 to 316 mV dec$^{-1}$, would be worse. Thus, this work investigates some electrical properties. First, both $N_{TA}$ and $N_{GA}$ negatively affect electrical performance under the MTR model. Second, the $N_{TA}$ significantly changes the $I_{on}/I_{off}$. However, the $N_{GA}$ significantly changes the positive shift $V_{th}$. If we employ the $N_{TA}$ in this study, the acceptor-like traps should increase around the end of the $E_C$. Therefore, the electrons should be trapped and released more easily by the $N_{TA}$ around the end of the $E_C$ when the device is operated. However, if we employ the $N_{GA}$ in this study, the acceptor-like traps increase around the $E_{GA}$ because $N_{GA}$ is not located at the edge of $E_C$. This means that electrons are trapped in acceptor-like states, but in the $N_{GA}$ case, electrons must have more energy to be emitted because they are not located near $E_C$ when the device is operated. Finally, other condition studies also run simulations of parameters, such as $W_{GA}$ and $E_{GA}$, with ranges of 0.05$\overline{W}$0.3 eV at $N_{GA} = 5 \times 10^{16}$ cm$^{-3}$ eV$^{-1}$ and 0.3–0.6 eV in $E_C$–$E$ at $N_{GA} = 5 \times 10^{16}$ cm$^{-3}$ eV$^{-1}$. However, these did not change the electrical performance. The results show that it is critical to reduce the amount of $N_{TA}$ and adjust the $E_{GA}$ control position for making high-performance AOS IGZO TFTs. We also simulated the $N_{TD}$ and $W_{TD}$ in the sub-gap DOS, ranging from $5 \times 10^{18}$ to $10^{20}$ cm$^{-3}$ eV$^{-1}$ and 0.03 to 0.06 eV at $N_{TD} = 10^{20}$ cm$^{-3}$ eV$^{-1}$. However, these did not change the electrical performance because, in this case, the IGZO channel layer is a large bandgap and $N_{TD}$ is far from the $E_C$ point. Therefore, the $N_{TD}$ near $E_V$ does not affect electrical performance when the device is operated. However, if high energy is injected, such as photons, it should release electrons in the states, and the electrons released from $N_{TD}$ to $E_C$ should change the electrical performance [25].

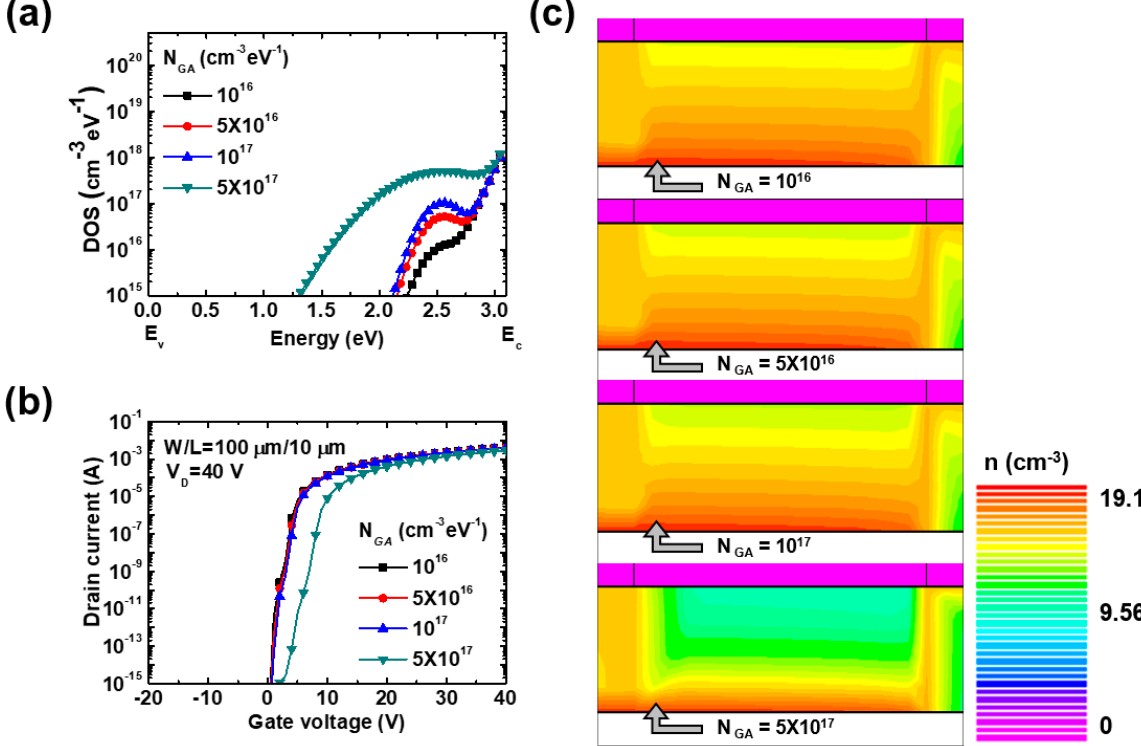

**Figure 3.** (**a**) DOS in sub-gap is shown as a function of energy. The Gauss donor-like states ($N_{GA}$) are controlled from $10^{16}$ to $5 \times 10^{17}$ cm$^{-3}$ eV$^{-1}$. (**b**) Drain current according to applied gate voltage is shown at 40 V of drain voltage. (**c**) Each layer depicts the electron concentration distribution from top to bottom of $N_{GA}$ ranging from $10^{16}$ to $5 \times 10^{17}$ cm$^{-3}$ eV$^{-1}$ at the saturation region 40 V of gate voltage and 40 V of drain voltage. The color table displays the electron concentration variations of 0–$10^{19.1}$ in varying colors.

Figure 4a shows the $N_{GD}$–$E$. The range of $N_{GD}$ is from $5 \times 10^{16}$ to $10^{18}$ cm$^{-3}$ eV$^{-1}$ by five intervals. Figure 4b shows the transfer curve changes with $N_{GD}$. It shows sweeping $V_G$ from −20 to 40 V at $V_D$ =

40 V of IGZO TFTs from each $N_{GA}$. Thus, the electrical performances, such as low $V_{on}$ from 0.85 to −3.4 V, and high *S* from 172 to 744 mV dec$^{-1}$, would be better and worse, respectively. Note that the distribution of $N_{GD}$ affects the electrical performance, such as a bad *S* and negative-shift $V_{th}$. Therefore, $N_{GD}$ is related to trapping electrons and associated with electron generation in the channel more easily by $V_O$. Therefore, if the $N_{GD}$ increases, *S* deteriorates, and $V_{th}$ changes in a negative shift. We elucidate more detailed reasons by examining the electron concentration distribution (Figure 4c). From the top to the bottom pictures, we find that the electrons gathered more easily on the channel by the condition $N_{GD}$ at the saturation region of the simulated IGZO TFTs $V_D = 40$ V and $V_G = 0$ V. Note that these results illustrate a tendency by the OV model. Therefore, the oxygen concentration must be controlled, and the $N_{GD}$ level via the PO and OV models must be adjusted.

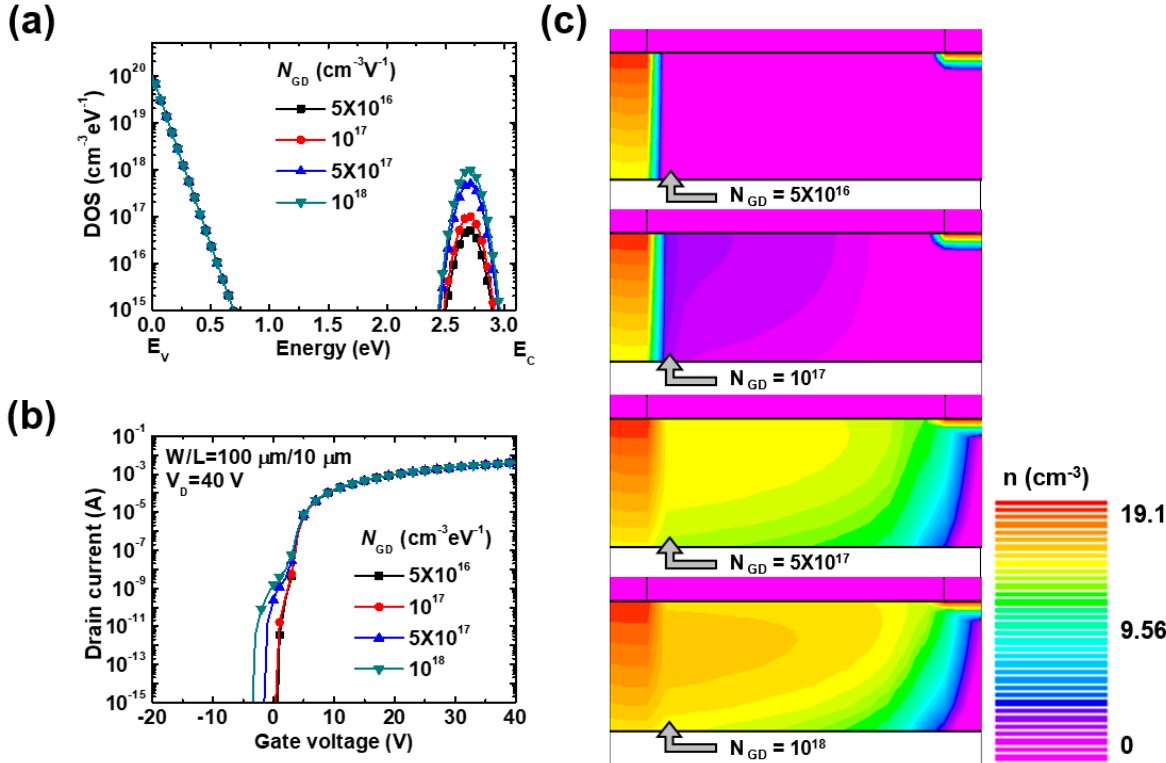

**Figure 4.** (**a**) DOS in sub-gap is shown as a function of energy. The Gauss acceptor-like states ($N_{GD}$) are controlled from $5 \times 10^{16}$ to $10^{18}$ cm$^{-3}$ eV$^{-1}$. (**b**) Drain current according to applied gate voltage is shown at 40 V of drain voltage. (**c**) Each layer depicts the electron concentration distribution from top to bottom of $N_{GD}$ ranging from $5 \times 10^{16}$ to $10^{18}$ cm$^{-3}$ eV$^{-1}$ at the saturation region 40 V of gate voltage and 40 V of drain voltage. The color table displays the electron concentration variations of $0$–$10^{19.1}$ in varying colors.

Figure 5a shows the $E_{GD}$–$E$. The range of $E_{GD}$ is from 2.3–2.9 eV for the $E_C$–$E$. Figure 5b shows that the transfer curve is changing with $E_{GD}$. It presents by sweeping $V_G$ −20 to 40 V at $V_D = 40$ V of IGZO TFTs from each $E_{GD}$. The electrical performances, such as high $I_{on}/I_{off}$ from $3.7 \times 10^{12}$ to $3.9 \times 10^{12}$ A and high $\mu_{sat}$ from 9.9 to 10.4 cm$^2$ V$^{-1}$·s$^{-1}$, low $V_{th}$ from 4.9 to 3.2 V, low $V_{on}$ from 0 to −2.2 V, and low *S* from 407 to 152 mV dec$^{-1}$, are shown. As the $E_{GD}$ approaches $E_C$, the electrical performances are improved, but the slope of the transfer curve changes around 0 V, showing a non-ideal transfer characteristic. Therefore, optimized electrical characteristic requires moving up by controlling the localized states in donor-like states by O–O bonding and $V_O$. For elucidating more detailed reasons, we investigated, as shown in Figure 5c. From the top to the bottom pictures, we find that the carrier concentration in the channel is high because $E_{GD}$ is located near $E_C$ more closely at $V_G = 0$ V and $V_D = 40$ V. Therefore, $E_{GD}$ is concerned with the carrier injection as a doping semiconductor by $V_O$.

Therefore, when $E_{GD}$ is located near $E_C$, electrons are generated more easily. Therefore, the electrons are released rather than trapped, and the electrical performance could be improved.

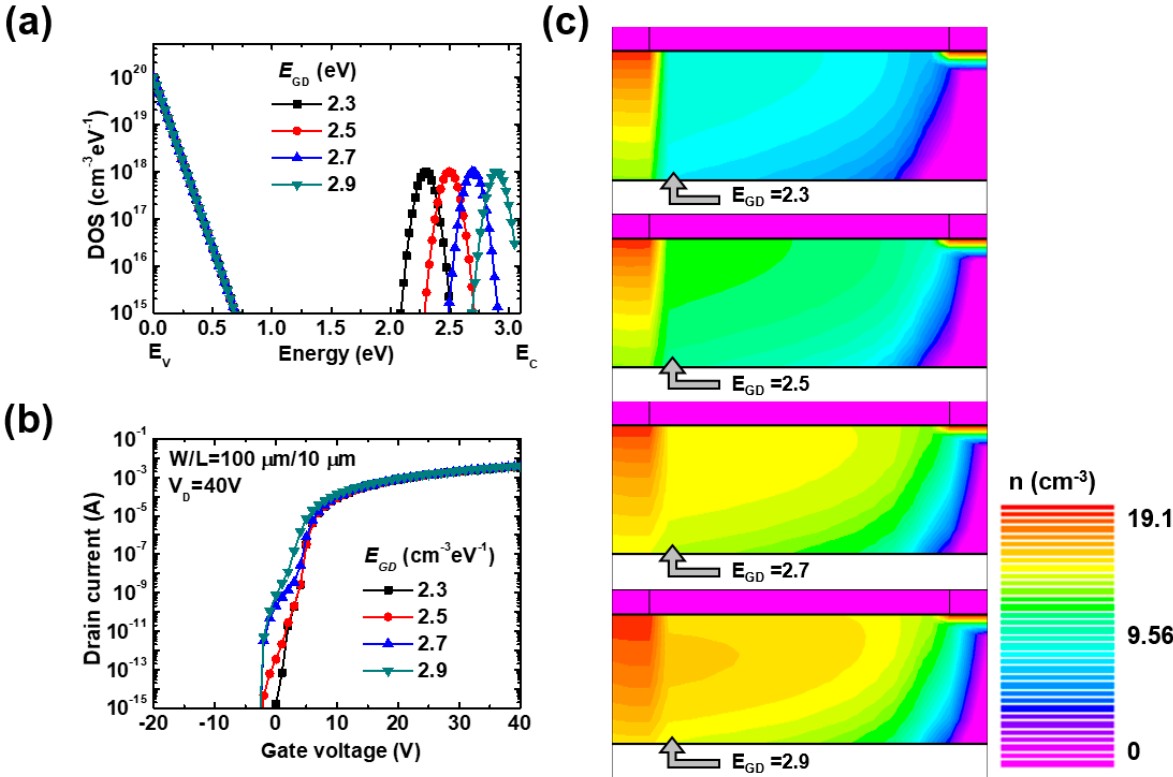

**Figure 5.** (**a**) DOS in sub-gap is shown as a function of energy. The $E_{GD}$ is controlled from 2.3–2.9. (**b**) Drain current according to applied gate voltage is shown at 40 V of drain voltage. (**c**) Each layer depicts the electron concentration distribution from top to bottom of $N_{GD}$ ranging from 2.3–2.9 eV at the saturation region 40 V of gate voltage and 40 V of drain voltage. The color table displays the electron concentration variations of 0–$10^{19.1}$ in varying colors.

## 4. Conclusions

We have described the importance of key parameters of acceptor-like and donor-like states in sub-gap states. From the simulation model and related theories, this work clarified how the sub-gap DOS profile affects the electrical characteristics in the a-IGZO TFTs. Note that four controlled variables, namely, $N_{TA}$, $N_{GA}$, $N_{GD}$, and $E_{GD}$, significantly affected electrical performances such as $V_{on}$, $V_{th}$, $S$, $I_{on}/I_{off}$, and $\mu_{sat}$. For acceptor-like traps, all electrical properties became worse because of the trapped electrons near the $E_C$ base on the MTR effect. For donor-like traps, some electrical properties such as $S$ became worse because of localized states. Other electrical properties such as $V_{th}$ and $\mu_{sat}$ improved because of the injected electrons based on the OV and PO models. In the future, we want to investigate the effect of more than four variables focused on in this work on electrical properties to exactly determine the significant role of controlling parameters on the improvement of the electrical properties in AOS TFTs.

**Author Contributions:** D.-K.K., J.-H.B. designed the research. D.-K.K. and J.P. (Jihwan Park) conducted simulation. D.-K.K., J.P. (Jihwan Park) and X.Z. analyzed the data and wrote the manuscript. J.P. (Jaehoon Park) and J.-H.B. edited the manuscript. All authors reviewed the manuscript. Project administration was conducted by J.P. (Jaehoon Park) and J.-H.B. All authors have read and agreed to the published version of the manuscript.

**Funding:** This research was funded by the Basic Science Research Program through the National Research Foundation of Korea (NRF) funded by the Ministry of Science and ICT (2018R1A2B6008815). The research was also supported by Hallym University Research Fund, 2020 (HRF-202009-015).

**Conflicts of Interest:** The authors declare no conflict of interest.

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
