# Peer review of "Numerical Study of Sub-Gap Density of States Dependent Electrical Characteristics in Amorphous In-Ga-Zn-O Thin-Film Transistors"

_electronics, doi:10.3390/electronics9101652_

Round 1

Reviewer 1 Report

The comments are enclosed.

Reviewer 2 Report

The article deals with the simulation of amorphous metal-oxide transistor characteristics in relation to a set of sub-bandgap states of varying concentrations. The study is useful because it provides insight into the impact of different types of defects on the transistor parameters. The discussion examines in detail the simulation results. Therefore, I recommend this paper for publication, provided that the following minor revisions are made

  1. , line 31: 'Metal-oxide thin-film transistors' should be changed into 'Thin-film transistors'. Perhaps the authors are referring to the fact that such transistors rely on a metal-insulator-semiconductor stack. However, using the phrase 'metal-oxide' in this context gives rise to ambiguity, because: a) not all insulators used in TFTs are oxides; b) the reader may misinterpret the phrase 'metal-oxide thin-film transistors' as referring only to TFTs based on amorphous metal-oxide semiconductors.

  1. 1, lines 31-32: the authors refer to TFTs as being relevant to active matrices only. This does not cover all cases, as TFTs are also being explored for applications in computing (e.g., DOI: 10.1038/s41586-019-1493-8), radiofrequency tags (e.g., DOI: 10.1109/ISSCC.2017.7870359), and analog signal processing (e.g., DOI: 10.1017/9781108559034). Therefore, the authors are advised to revise this statement by including all these additional cases and the respective references recommended.

  1. The trap concentration values provided in the figure captions are formatted incorrectly. Please revise.

Reviewer 3 Report

This paper study the sub-gap density of states of IGZO TFTs by numerical analysis. The authors demonstrate the effect of oxygen vacancy, peroxide, hydrogen complex, band-to-band tunneling, and trap-assisted tunneling on the device performance. This paper is well writen and can be published as it is.

Author Response

As the reviewer mentioned, we have shown the effect of the sub-gap density of states on device performance.

We carefully checked for typos and minor errors in the manuscript.

We appreciate the reviewers' kind reviews.